# A Moderated Mediation Model of the Relationship between Family Dynamics and Sleep Quality in College Students: The Role of Big Five Personality and Only-Child Status

**DOI:** 10.3390/ijerph19063576

**Published:** 2022-03-17

**Authors:** Xiaocen Jia, Yiqing Huang, Wenli Yu, Wai-Kit Ming, Fei Qi, Yibo Wu

**Affiliations:** 1School of Public Health, Qingdao University, Qingdao 266071, China; jiaxiaocen1996@163.com (X.J.); huangyiqing96@163.com (Y.H.); 2School of Liberal Education, Weifang University of Science and Technology, Shouguang 262700, China; y18754409860@163.com; 3Department of Infectious Diseases and Public Health, Jockey Club College of Veterinary Medicine and Life Sciences, City University of Hong Kong, Hong Kong 999077, China; wkming2@cityu.edu.hk; 4Qingdao Municipal Center for Disease Control and Prevention, Qingdao 266033, China; 5Qingdao Institute of Preventive Medicine, Qingdao 266033, China; 6School of Public Health, Peking University, Beijing 100191, China; 7Health Culture Research Center of Shaanxi, Key Research Base of Philosophy and Social Sciences in Shaanxi Province, Xi’an 725106, China

**Keywords:** family dynamics, agreeableness, conscientiousness, quality of sleep, only-child status, moderated mediation, Chinese undergraduate students

## Abstract

Sleep quality among college students is affected by numerous factors. Previous studies have linked sleep quality to family dynamics as well as personality. However, little is known about the mechanisms underlying this relationship. The aim of this study is to incorporate a moderated mediation model to explore the big five personality traits in mediating the relationship between family dynamics and sleep quality and whether this indirect relationship is modified by only-child status among Chinese undergraduate students. Survey data were collected from a cross-sectional study conducted in Shandong, China and 1022 undergraduate students aged 18–24 were recruited. The mediation and moderated mediation modeling analyses were carried out with the software SPSS PROCESS macro. After controlling for gender and grade, mediation analysis indicated that conscientiousness and agreeableness of the big five personality traits partially mediated the link between family dynamics and sleep quality (β = −0.0093, CI: −0.0178, −0.0018; β = −0.0047, CI: −0.0084, −0.0013), and moderation analysis found only-child status acted as a moderator in the relationship between family dynamics and the agreeableness of the big five personality traits (only child, β = −0.0129, CI: −0.0196, −0.0072; non-only-child, β = −0.0040, CI: −0.0078, −0.0002). Results showed that family dynamics positively predicted sleep quality both directly and indirectly via the conscientiousness and agreeableness of big five personality traits. Only-child status moderated the indirect association between family dynamics and agreeableness of big five personality traits. The current study advanced our understanding of the mechanism underlying the connection between family dynamics and sleep quality and helped to develop intervention measures to improve sleep quality of college students.

## 1. Introduction

Over the past few decades, the Big Five personality structure model has been extensively studied and proved to have cross-assessor and cross-cultural stability. It has been widely accepted at the dimensional level by personality psychologists. It has been the most influential human ethics model around the world [1]. Five personality traits have been formed in recent decades, including neuroticism, conscientiousness, agreeableness, extraversion, and openness [2]. Neurotic personality is defined as anxiety, self-pity, depression, and impulsivity [3]. Conscientious personality is characterized by neatness, self-discipline, organization, and dependability [4]. Agreeable personality is trust, tolerance, kindness, compassion, humility, and attaches importance to getting along with others [3]. Extraversion reflects the extent to which one is talkative, social, gregarious, and assertive [2]. The last of the five personalities is openness; an open personality may be artistic (in the fields of music, art, and poetry), highly curious, imaginative, and insightful [3]. Personality development is a stable and constantly improving process. Individual genetic factors determine the nature of personality. At the same time, the acquired interpersonal processes and experiences shape and improve individual personality constantly, of which the influence of family is the most significant [5]. Family is the earliest and longest living environment, which plays a significant role in the influence of college students’ personality traits.

Family has a significant impact on individual growth and physical and mental health. In the research of family system, it is usually studied in two ways. One is to explain the family system through family function, the other is to use the family dynamic characteristics to measure the internal characteristics of the family system. Family dynamics emphasizes the dynamic process of family internal interaction and can present a unique method of interaction among family members [6]. The study of family dynamics began in the 1950s and developed along with the study of family therapy [7]. Family dynamics is a description of the characteristics of family system. Compared with research tools such as the Family Environment Scale, which are more commonly used in domestic family research, this description emphasizes a dynamic interactive process [8]. The structure and organization of the family are extremely important factors that influence and determine the behavior of family members. In China, the only-child policy was launched in 1980 [9], which restricted urban families to one child while rural families, minority families, and blended families could have up to two. It is generally believed that only-child groups and non-only-child groups have different personality, cognition, and influence characteristics due to the influence of family environment [10]. Only-children receive too much attention and excessive praise from their parents and grandparents, which may lead to undesirable personality traits, such as selfishness, dependency, and social incompetence [11].

College students are important reserve talents for national construction, and it has always been the goal of society to promote their physical and mental health. College students’ sleep quality is closely associated with their health and personal development [12]. A good sleep is an important basis for maintaining the physical and mental health, as well as the complete social functions of an individual [13]. In the latest interpretation of the health standards of the United Nations health organization, “good sleep quality” is included, because sleep quality problems affect physical, psychological, and social health; the incidence of diseases; the generation of problematic emotions or behaviors; and the development of social dysfunctions [14]. It is shown that 18.7–21.4% of college students have sleep problems in China [15], such as difficulty falling asleep and low sleep efficiency, which affect the study and activities of the following day seriously. A survey shows that the quality of sleep among Chinese college students is deteriorating [16]. Consequently, it is important to study the sleep quality of college students.

There are various factors influencing the sleep quality of college students. Numerous studies have explored the relationship between family-related factors and sleep quality, as well as the relationship between Big Five personality and sleep quality. In terms of family-related factors, studies have shown that only-child status, family economic status, and mother’s education level are related to the sleep quality of college students [17]. In the area of personality traits, study on the relationship between neuroticism and sleep quality is the most in-depth, and the results show that neuroticism is consistently, significantly negatively correlated with sleep quality, which is a strong risk predictor of sleep quality [18]. A survey shows that among young Korean women, neuroticism is the strongest personality factor affecting sleep quality and might be the best predictor for sleep quality, while conscientiousness is the best predictor of poor sleep quality status [19]. Compared with neuroticism dimension, the research on the relationship between other dimensions and sleep quality is not rich enough and the research results are inconsistent. However, few studies have combined family-related factors, especially family dynamics, with the Big Five personality to explore the impact on college students’ sleep quality.

To fill these gaps, the present study built a moderated mediation model to explore the mediation and the moderation effects of big five personality traits in the association between family dynamics and sleep quality. The purpose of this study is twofold. First, it aimed to examine the mediating roles of big five personality traits in the association between family dynamics and sleep quality. We hypothesized that all aspects of the big five personality would significantly mediate this relationship. Second, this study explored the moderating effect of only-child status. The hypothesized model is shown in Figure 1.

## 2. Materials and Methods

### 2.1. Participants and Procedures

The survey was conducted in Shandong province, China from 20 March to 31 March 2021. Stratified cluster sampling was used to ensure the diversity of participants. According to the geographical distribution characteristics of Shandong Province, two universities were selected from the four regions of West Shandong, South Shandong, Central Shandong, and East Shandong. An online questionnaire was adopted in this survey. Investigators were recruited and standardized training was given to the investigators. Digital questionnaires were established and saved online. Investigators excluded participants who had long-term use of sedative-hypnotic drugs such as diazepam as well as other sleep disorders through questioning and then forwarded to participants by investigators through the link of the “Wenjuanxing” widely used in China. The survey was conducted at the end of each class. Participants clicked the link and received the questionnaires, which took approximately 10–20 min to complete. In the end, questionnaires were collected automatically when participants clicked the submit button. A total of 1022 questionnaires were collected and 963 were valid for analyses. Sociodemographic characteristics information included Gender, Ethnicity, Grade, Major, Location of their home, Whether only-child status, and Family type (see Table 1). The survey protocols, instruments, and the process for obtaining the informed consent for participants were carried out in accordance with relevant guidelines and regulations and were approved by the Ethics Committee of Shaanxi Health Culture Research Center (protocol code JKWH-2021-03 and 2021.01). In order to stimulate participation, every participant received a small gift after completing the questionnaire.

### 2.2. Measures

#### 2.2.1. Self-Rating Scale of Systemic Family Dynamics (SSFD)

The SSFD [6] consists of 23 items, which are composed of four dimensions: family atmosphere, individuation, system logic, and disease concept. Family atmosphere consists of 8 items such as “our family members have deep feelings for each other, and when one of them is in trouble, everyone will feel pain and anxiety”, which refers to the emotional characteristics of communication within the family system. Individuation consists of 6 items such as “our family allows family members to live in their own way”, which is used to evaluate the degree of emotional and behavioral differentiation among family members. System logic consists of 5 items such as “our family does not like people who have different views with us”, which is used to evaluate the logical characteristics of family members’ value judgments. The disease concept consists of 4 items such as ”people in our family believe that self-adjustment of mental state can treat mental illness”, which refers to the family members’ self-responsibility for the disease process. Participants respond to items on a five-point scale, ranging from 1 (“does not describe me well”) to 5 (“describe me very well”). The total score ranges from 23 to 115. The higher the score, the better the family system of the subject. The Cronbach’s alpha in this research was 0.910, indicating high internal consistency.

#### 2.2.2. Chinese Big Five Personality Inventory Brief Version (CBF-PI-B)

Personality traits were assessed with the Chinese Big Five Personality Inventory brief version (CBF-PI-B) [20], including neuroticism, conscientiousness, agreeableness, extraversion, and openness. The CBF-PI-B consists of 40 items with a 0 (strongly disagree) to 6 (strongly agree) Likert format. Each trait score is derived from 8 items, with all trait scores ranging from 0 to 48. The higher the total score of each trait, the higher the degree of the individual’s personality trait. The Cronbach’s alpha in this research were obtained as follows: neuroticism, 0.774; conscientiousness, 0.841; agreeableness, 0.635; extraversion, 0.728; openness, 0.927; these indicate that all facets had fair reliabilities.

#### 2.2.3. Pittsburgh Sleep Quality Index (PSQI)

The Pittsburgh Sleep Quality Index (PSQI) [21] is a self-assessment questionnaire used to assess participants’ sleep quality and sleep disorders in the last month. The PSQI contains 19 items that can be grouped into seven components: subjective sleep quality, sleep latency, sleep duration, habitual sleep efficiency, sleep disturbances, use of sleeping medication, and daytime dysfunction. The score for each component ranges from 0 to 3 points. The PSQI total score ranges from 0 to 21. Lower scores indicate better sleep quality. It is shown that the PSQI has high validity and reliability and is a relatively universal sleep quality screening tool at present [22]. The PSQI has been proven to be reliable and valid in China [23]. The Cronbach’s alpha in this research was 0.920.

### 2.3. Data Analyses

#### 2.3.1. Statistical Analyses

In this study, SPSS 26.0 (SPSS, Chicago, IL, USA) was used for statistical analyses and the significance level was set at 0.05. First, the Harman single-factor test was used to perform factor analysis on all variables combined in the questionnaire. The results of this study showed that 24.36% of the variation was explained by the first principal component, which was below the critical value (40%), indicating that there was no common method bias effect among the variables measured. Second, the data distribution is examined and found to be non-normal. According to Preacher et al. [24] and Hayes [25], the analyses via the SPSS PROCESS macro had no requirement on the distribution of the data. The SPSS PROCESS macro is based on the Bootstrapping test. Bootstrapping is one of several resampling strategies for estimation and hypothesis testing. In bootstrapping, the sample is conceptualized as a pseudo-population that represents the broader population from which the sample was derived and the sampling distribution of any statistic can be generated by calculating the statistic of interest in multiple resamples of the data set. Using bootstrapping, no assumptions about the shape of the sampling distribution of the statistic are necessary when conducting inferential tests [24]. In this study, Spearman rho correlations were used to check the correlations of all variables. Third, model 4 of SPSS macro PROCESS3.1 was used to test the mediating roles of big five personality in the relationship between family dynamics and sleep quality. Using the method and software described in model 4, model 7 was conducted to test the moderated mediation model. In addition, bias-corrected bootstrapping procedures with 5000 resamples were utilized to calculate 95% confidence intervals of the direct and indirect effects.

#### 2.3.2. Control Variables

First of all, through literature review, we found differences in sleep quality of college students by gender [26] and grade [27], while no differences were found in other demographic variables. Furthermore, Spearman rho correlation results showed that gender and grade were significantly correlated with the investigated variables (see Table 2); thus, they were treated as control variables in all analyses afterward. Gender was coded as males 1 and females 2. Freshmen were coded as 1 and sophomores and above were coded as 2.

## 3. Results

### 3.1. Preliminary Analyses

The relationship between family dynamics, big five personality traits, PSQ, and whether only child is shown in Table 2. Family dynamics is positively correlated with conscientiousness, agreeableness, openness, and extroversion among the big five personality traits. Neuroticism of the big five personality traits was significantly positively associated with PSQI. Conscientiousness, agreeableness, and extraversion among the big five personality traits was significantly negatively related to PSQI. Family dynamics was significantly negatively associated with PSQI (see Table 2).

### 3.2. Testing for the Mediation Effects of Big Five Personality

As showed in Table 3 and Table 4, the results of the mediation analyses showed that family dynamics negatively predicted PSQI both directly (β = −0.0552, *p* < 0.001) (see Table 3) and indirectly via conscientiousness and agreeableness of the big five personality traits (β = −0.0093, CI: −0.0178, −0.0018; β = −0.0047, CI: −0.0084, −0.0013) (see Table 4) after controlling for gender and age. Namely, conscientiousness and agreeableness of the big five personality traits partially mediated the impact of family dynamics on PSQI. There was no significant mediating effects observed for the other three detected paths (neuroticism: β = 0.0050, CI: −0.0023, 0.0121; openness: β = 0.0048, CI: −0.0053, 0.0148; extraversion: β = 0.0053, CI: −0.0014, 0.0128) (see Table 4).

### 3.3. Testing for the Moderated Mediation Effects of Only-Child Status

The test results of SPSS macro PROCESS 3.1 on the moderated mediation effects are shown in Table 5 and Table 6. After controlling for gender, grade, family dynamics negatively predicted PSQI both directly (β = −0.0468, *p* < 0.001) (see Table 5) and indirectly via the agreeableness of the big five personality traits (only child, β = −0.0129, CI: −0.0196, −0.0072; non-only-child, β = −0.0040, CI: −0.0078, −0.0002), not via conscientiousness (only child, β = 0.0004, CI: −0.0130, 0.0117; non-only-child, β = 0.0002, CI: −0.0052, 0.0049) (see Table 6). Namely, the agreeableness of the big five personality trait partially mediated the relationship between family dynamics and PSQI. The interaction of family dynamics and whether only child had a significant effect on agreeableness: (family dynamics × whether only child, β = −0.0677, *p* < 0.001, CI: −0.1047, −0.0309) (see Table 5). The verified moderated mediation model is shown in Figure 2. For the further understanding of the moderation effect of only-child status on the path between family dynamics and agreeableness, a conditional indirect effect analysis was conducted at two aspects of only-child status in Figure 3. There is a significant positive correlation between family dynamics and agreeableness in the only-child family, but not in the non-only-child family. The slope illustrates that the positive effect of family dynamics on agreeableness was greater for the only-child family.

## 4. Discussion

Our investigation focused on family dynamics, sleep quality, and big five personality traits. By constructing a moderated mediation model, we found that family dynamics positively predicted sleep quality both directly and indirectly via the conscientiousness and agreeableness of big five personality traits. In addition, only-child status moderated the indirect association between family dynamics and agreeableness of big five personality traits.

First of all, a preliminary correlation analysis was performed on all variables in this study. It is interesting that family dynamics was positively associated with all the big five personality traits except neuroticism. Familial influences on neuroticism and education in the UK Biobank showed that no effect of family environment on neuroticism was found, which is consistent with the findings of the present study [28]. Another possible explanation is that neuroticism is a vulnerability factor, while the remaining traits are protective factors [29]. Families with better systematic family dynamics are moderately cohesive and adaptable, with both family structure and flexibility, providing boundaries for family members and freedom for growth and development. Therefore, college students from families with better systematic family dynamics were more conducive to the formation of the protective factor personality. Neuroticism was significantly positively associated with PSQI score. Scoring higher on neuroticism was related to worse sleep quality. Neuroticism represents a tendency to experience anxiety and pain and is usually related to excessive negative cognitive activities (worry and contemplation). Cognitive processes, especially the inability to shut down or control thoughts, are considered to be an important influencing factor in insomnia [30,31]. In addition, hostility is an aspect of neuroticism and is associated with poor sleep quality [32]. Individuals with higher neuroticism are more sensitive to stressors [33], which may amplify and prolong sleeping difficulties at the same time. Conscientiousness, extraversion, and agreeableness among the big five personality traits were significantly negatively associated with PSQI. A high degree of conscientiousness is characterized by a sense of responsibility and organization, which is associated with physical health, longevity, and good coping ability in the face of difficulties [34,35]. These qualities can help those with a strong sense of conscientiousness develop good sleep hygiene habits so as to ensure good sleep. Extraversion is associated with a lower stress response and a more physically active lifestyle [36], which may help to reduce the possibility of sleep difficulties. More agreeable individuals also had lower levels of stress [37] and better sleep quality.

Secondly, consistent with our hypothesis, this study indicates that family dynamics has a positive impact on sleep quality among college students. The study of family dynamics originated from Systemic Family Therapy (SFT), which is one of the branches of family therapy that has developed rapidly in the past 40 years. It is characterized by the use of system theory, cybernetics, information theory, and game theory to explain family structure and planning treatment techniques [38]. With the development of family dynamics theory research, the family dynamics theory system has become increasingly abundant; in conclusion, it can reflect the family situation more comprehensively and systematically [8]. Family is an important source of support and emotional security. Close interpersonal relationships can cultivate a sense of support and security and can influence mood and anxiety positively, which may potentially affect sleep [39]. In addition, families with better family dynamics have less control over their children by parents, allowing their children to have their own independent development space [6], so there is relatively less pressure from family. College students may experience feelings of stability and safety, which are protective against poor sleep.

Furthermore, it is indicated that conscientiousness and agreeableness among the big five personality traits partially mediated the impact of family dynamics on sleep quality, which partially supported our hypotheses. Family dynamics have a very important impact on the mental health and personality development of college students [40]. The higher the score of family dynamics, the more likely it is to form a secure personality, such as conscientiousness and agreeableness. Furthermore, conscientiousness is associated with physical activity [36], lower body mass index [41], and lower likelihood of smoking [42], which may mediate the relationship between high conscientiousness and better sleep quality. A previous study showed that conscientiousness is related to fewer dangerous health behaviors and more health-promoting behaviors, including less alcohol and drug use, unhealthy eating habits, dangerous sex behaviors, dangerous driving, tobacco use, suicide, and violent behavior [43]. Extending to conscientious personality can also predict healthy behaviors that promote sleep. Agreeableness was found to be an important predictor of sleep quality, which is consistent with previous findings and plays a parallel mediating role along with conscientiousness in this study. In general, high agreeableness seems to be associated with good and adequate sleep and those who are more agreeable may be better at following sleep-related advice [37]. Less stress in more agreeable people may also be an explanation [44]. However, conscientiousness and agreeableness of the big five personality traits only partially mediates the relationship between family dynamics and sleep quality, and the mediating effect was less than 1%, implying that personality traits account for only a small portion of the effect of family dynamics on sleep quality. However, given the prevalence of sleep quality problems among college students, even small improvements could have a considerable public health impact.

Finally, this study proved the moderation effect of only-child status on the indirect association between family dynamics and agreeableness. In other words, the indirect effect of family dynamics on agreeableness from an only-child family was stronger compared with those from non-only-child family. This finding was in line with the viewpoint that there is a difference in agreeableness between the only-child group and the non-only-child group [11]. Children who are only-child’s receive their families’ undivided attention, unlike the siblings who share their families’ attention [45]. In families with an only-child, attention, time, and energy provided by parents of the only-child may lead to better parental guidance and individual care [46]. Thus, an only-child may adjust better psychologically and behaviorally, which may be more conducive to the formation of agreeable personality. Moreover, only-children in China usually live in more economically developed areas (e.g., Chinese registered residences—known as “Hukou”—in rural areas were legally permitted to have a second child if their first one was female); have parents with higher education level and better occupational background; and have a richer and more diverse extracurricular life; thus, they have a more pleasant childhood [47]. These factors contribute to the formation of their agreeable personality. Therefore, the family dynamics of college students from only-child families have a greater impact on the agreeableness personality. Overall, this study firstly integrated the only-child as a moderator into the trait family dynamics–sleep quality model. On a theoretical level, it deepened our understanding of the individual differences in this process; as for the practical level, we offered new directions for improving the sleep quality of college students purposively.

### Limitations and Implications

There are still some limitations that need to be resolved in future studies. First, the current cross-sectional study cannot determine the causal relationship between family dynamics and sleep quality. Future research could adopt a longitudinal design or experiments to explore the causal relationship between family dynamics and sleep quality through aggregating cross-sectional design and a multilevel linear model or manipulation of independent variables and intermediary variables. Second, only Chinese undergraduates were included in this study. The results can only be extended to the age group of 18–24 years old and cannot represent the entire population. Future research should consider a much wider age range of the population to see if this pattern of results can replicate in non-college-student samples as well.

Despite limitations, the results of the current study have important practical implications. One of the important research implications of this study may be that it is the first study to examine the mechanism among family dynamics, big five personality, and sleep quality. Family dynamics could positively predict sleep quality directly and indirectly via conscientiousness and agreeableness of the big five personality traits and whether only child moderated the indirect relationship between agreeableness and sleep quality. Examining personality traits may help build a knowledge base that can be used to map phenotypes associated with individual sleep differences and identifying sleep-related phenotypes may help to examine the underlying genetic background and physiological mechanisms [48].

## 5. Conclusions

This study involved 963 Chinese undergraduates and examined the mediating role of the big five personality in the relationship between family dynamics and sleep quality, and the moderating role of only-child status. Based on our findings, it is helpful to develop intervention measures to improve sleep quality of college students, and by improving sleep quality, it can also improve academic performance and overall health of college students.

## Figures and Tables

**Figure 1 ijerph-19-03576-f001:**
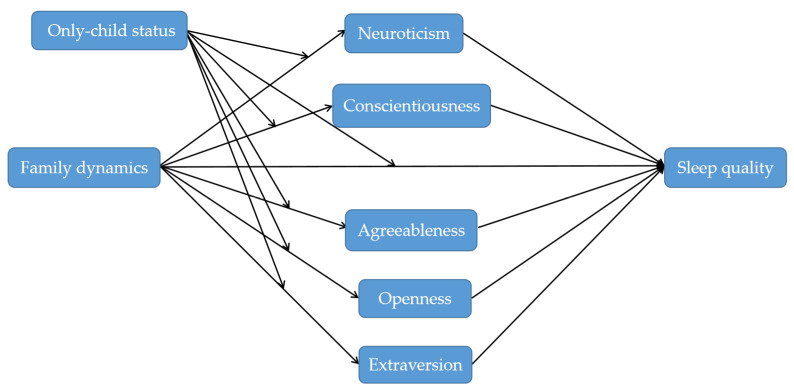
The hypothesized model.

**Figure 2 ijerph-19-03576-f002:**
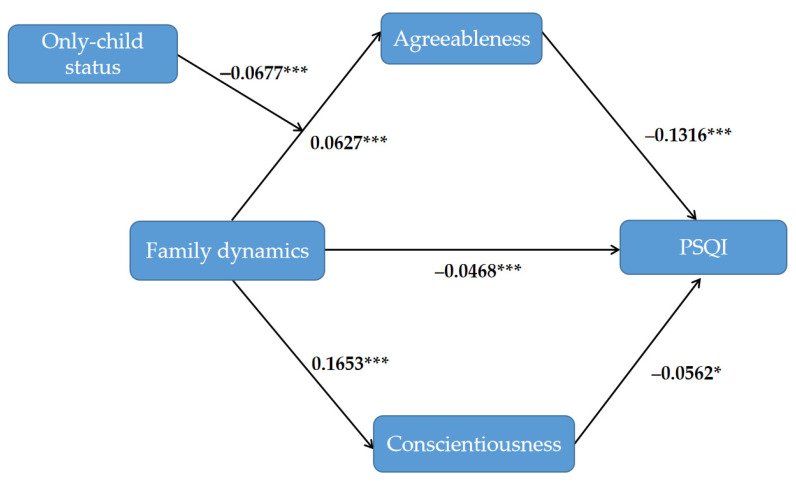
The verified model of the moderated mediation effects between family dynamics and PSQI (N = 963). The study found that conscientiousness and agreeableness of big five personality traits mediated the association between family dynamics and sleep quality. Only-child status significantly moderated the impact of family dynamics on agreeableness.* *p* < 0.05, *** *p* < 0.001.

**Figure 3 ijerph-19-03576-f003:**
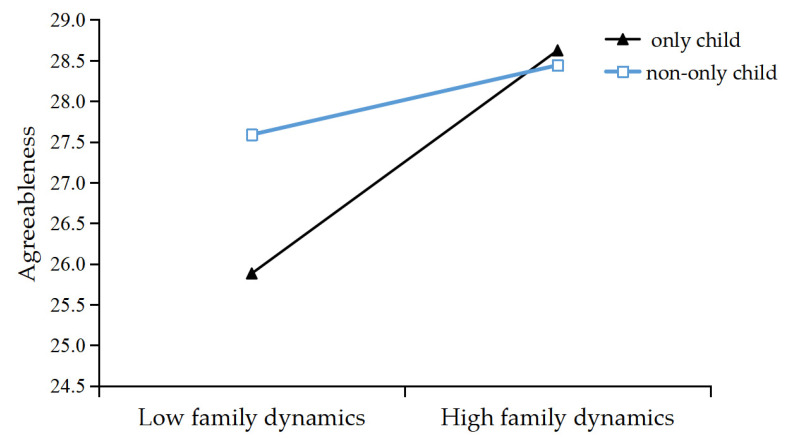
The conditional effect between family dynamics and agreeableness (N = 963). To further understand the moderation effect of only-child status on the path between family dynamics and agreeableness, a conditional indirect effect analysis was conducted at two aspects of the only-child and the non-only-child.

**Table 1 ijerph-19-03576-t001:** Summary of the demographic variables (N = 963).

Variables	Number	Percent (%)
Gender		
male	406	42.2
female	557	57.8
Ethnicity		
Han	855	88.8
Minorities	108	11.2
Grade		
Freshman	204	21.2
Sophomore	275	28.6
Junior	237	24.6
Senior	124	12.9
Fifth	123	12.8
Major		
Philosophy	36	3.7
Economics	99	10.3
Jurisprudence	50	5.2
Education	53	5.5
Literature	93	9.7
History	19	2.0
Science	73	7.6
Engineering	146	15.2
Agronomy	21	2.2
Medical	241	25.0
Management	90	9.3
Art	42	4.4
Location of their home		
Urban	516	53.6
Rural	447	46.4
Whether only-child status		
Only child	457	47.5
Non-only-child	506	52.5
Family types		
The nuclear family	610	63.3
Three generations	254	26.4
Four generations	38	3.9
Single parent familiy	43	4.5
Intergenerational family	18	1.9
Total	963	100

**Table 2 ijerph-19-03576-t002:** Key variables and Spearman correlation coefficients of all variables.

Variable	M ± SD	1	2	3	4	5	6	7	8	9
1. Family dynamics	80.90 ± 13.96	1.000								
2. Neuroticism	23.07 ± 7.03	0.005	1.000							
3. Conscientiousness	28.47 ± 5.66	0.399 ***	−0.005	1.000						
4. Agreeableness	27.65 ± 4.66	0.201 ***	−0.107 ***	0.577 ***	1.000					
5. Openness	28.75 ± 6.65	0.403 ***	0.169 ***	0.613 ***	0.416 ***	1.000				
6. Extraversion	25.98 ± 4.93	0.372 ***	0.024	0.497 ***	0.262 ***	0.642 ***	1.000			
7. PSQI	5.26 ± 3.77	−0.255 ***	0.376 ***	−0.163 ***	−0.192 ***	−0.043	−0.070 *	1.000		
8. Gender		−0.029	0.028	0.008	0.124 ***	−0.021	−0.057 *	0.012	1.000	
9. Grade		0.014	−0.079 *	0.026	0.083 **	−0.065 *	−0.087 **	−0.013	0.033	1.000

Notes: N = 963; * *p* < 0.05, ** *p* < 0.01, *** *p* < 0.001; PSQI, Pittsburgh sleep quality index.

**Table 3 ijerph-19-03576-t003:** Mediation analysis.

Outcome Variable	Independent Variables	β	*p*
PSQI	constant	9.8312 ***	0.0000
	gender	0.0974	0.6861
	grade	−0.0987	0.2842
	family dynamics	−0.0552 ***	0.0000
	R^2^	0.2081 ***	
	F	14.4769	
Neuroticism	constant	21.3890 ***	0.0000
	gender	0.5541	0.2257
	grade	−0.4849 **	0.0056
	family dynamics	0.0260	0.1082
	R^2^	0.1086 **	
	F	3.8116	
Conscientiousness	constant	14.7199 ***	0.0000
	gender	0.0709	0.8339
	grade	0.1008	0.4356
	family dynamics	0.1653 ***	0.0000
	R^2^	0.4082 ***	
	F	63.9210	
Agreeableness	constant	20.2823 ***	0.0000
	gender	0.9691 ***	0.0006
	grade	0.327 **	0.0026
	family dynamics	0.0613 ***	0.0000
	R^2^	0.2412 ***	
	F	19.7475	
Openness	constant	12.9850 ***	0.0000
	gender	−0.0838	0.8296
	grade	−0.3980 **	0.0076
	family dynamics	0.2097 ***	0.0000
	R^2^	0.4458 ***	
	F	79.2667	
Extraversion	constant	17.209 ***	0.0000
	gender	−0.4213	0.1574
	grade	−0.4082 ***	0.0004
	family dynamics	0.1301 ***	0.0000
	R^2^	0.3845 ***	
	F	55.4622	
PSQI	constant	7.8649 ***	0.0000
	gender	0.0878	0.6948
	grade	0.0505	0.5583
	neuroticism	0.1920 ***	0.0000
	conscientiousness	−0.0562 *	0.0480
	agreeableness	−0.0762 *	0.0123
	openness	0.0227	0.3905
	extraversion	0.0404	0.1692
	family dynamics	−0.0563 ***	0.0000
	R^2^	0.4449 ***	
	F	29.4289	

Notes: N = 963; * *p* < 0.05, ** *p* < 0.01, *** *p* < 0.001; PSQI, Pittsburgh sleep quality index.

**Table 4 ijerph-19-03576-t004:** Bootstrapping indirect effect and 95% confidence interval (CI) for the mediation test.

Indirect Path	Effect	BootLLCI	BootULCI
family dynamics→neuroticism→PSQI	0.0050 ^b^	−0.0023	0.0121
family dynamics→conscientiousness→PSQI	−0.0093 ^a^	−0.0178	−0.0018
family dynamics→agreeableness→PSQI	−0.0047 ^a^	−0.0084	−0.0013
family dynamics→openness→PSQI	0.0048 ^b^	−0.0053	0.0148
family dynamics→extraversion→PSQI	0.0053 ^b^	−0.0014	0.0128

Notes: N = 963; Bootstrap sample size = 5000; PSQI, Pittsburgh sleep quality index; LL, low limit; CI, confidence interval; UL, upper limit; ^a^ Empirical 95% confidence interval does not overlap with zero; ^b^ Empirical 95% confidence interval overlaps with zero.

**Table 5 ijerph-19-03576-t005:** The moderated mediation analyses.

Outcome Variable	Independent Variables	β	*p*
Conscientiousness	constant	28.0099 ***	0.0000
	gender	0.0694	0.8349
	grade	0.1374	0.2786
	family dynamics	0.1689 ***	0.0000
	only-child status	0.6522	0.0477
	family dynamics * only-child status	−0.1482 ***	0.0000
	R^2^	0.4504 ***	
	F	48.7205	
Agreeableness	constant	25.2655 ***	0.0000
	gender	0.9219 **	0.0011
	grade	0.3487 **	0.0012
	family dynamics	0.0627 ***	0.0000
	only-child status	0.7621 **	0.0064
	family dynamics * only-child status	−0.0677 ***	0.0007
	R2	0.2766 ***	
	F	15.8615	
PSQI	constant	8.7375 ***	0.0000
	gender	0.2250	0.3488
	grade	−0.0555	0.5444
	family dynamics	−0.0468 ***	0.0000
	conscientiousness	−0.0018	0.9456
	agreeableness	−0.1316 ***	0.0000
	R2	0.2580 ***	
	F	13.6497	

Notes: N = 963; * *p* < 0.05, ** *p* < 0.01, *** *p* < 0.001; PSQI, Pittsburgh sleep quality index.

**Table 6 ijerph-19-03576-t006:** Bootstrapping the conditional indirect effect and 95% confidence interval (CI) for the moderated mediation model.

Path	Only-Child Status	Effect	BootLLCI	BootULCI
family dynamics→conscientiousness→PSQI	only-child	0.0004	−0.0130	0.0117
	non-only-child	0.0002	−0.0052	0.0049
family dynamics→agreeableness→PSQI	only-child	−0.0129 ^a^	−0.0196	−0.0072
	non-only-child	−0.0040 ^a^	−0.0078	−0.0002

Notes: N = 963; Bootstrap sample size = 5000; PSQI, Pittsburgh sleep quality index; LL, low limit; CI, confidence interval; UL, upper limit. ^a^ Empirical 95% confidence interval does not overlap with zero.

## Data Availability

Data are available, upon reasonable request, by emailing: bjmuwuyibo@outlook.com.

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
