# Peer review of "A Moderated Mediation Model of the Relationship between Family Dynamics and Sleep Quality in College Students: The Role of Big Five Personality and Only-Child Status"

_ijerph, 2022, doi:10.3390/ijerph19063576_

Round 1
Reviewer 1 Report
Authors studied the mediating effect of Big Five Personality between family dynamics and sleep quality, by also taking into account the factor “only child”. Authors found that some factors of Big Five (conscientiousness, agreeableness) mediated the relationship between family dynamics and sleep quality; moderation analysis found that the factor only-child moderated the relationship between family dynamics and the agreeableness of the big five personality. This article shed light on the role of stable personality trait in mediating the biopsychological relationship between sleep quality and family dynamics, thus constituting a very interesting and innovative topic.
However, major revisions are needed to significantly improve the quality of the paper.
Title
The title is not clearly informative, as it not describes the direction of the preliminary evaluation that authors conducted. Title should also anticipate the results of the main outcomes to be “captivating”
Abstract:
Lines 17-18: This sentence is too vague, and does not improve the comprehension of the state of art/background that authors want to communicate: please, rephrase it or remove it. I appreciate authors reported the results of mediation and moderation analyses, but it is also important to let the interpretation of the results emerge; In. other words, authors should also spend few sentences to discuss the results obtained. In this way, the abstract could be “stand-alone”, thus providing essential and pivotal information to the readers.
Keywords: I suggest authors to find synonyms for the keywords that are also present in the title: this alternative might offer more visibility to the current work.
Introduction
The introduction section is too compartmentalized. I suggest to create a narrative link between the major components of investigation: in other words, I think that the entire introduction should be rethought in a way to deal with the rationale of the study. For example, authors could start with the description of personality traits and then, following that so called funnel-approach, they could explain the association of personality traits, family dynamics, sleep quality and being only child in a more fluid way. I appreciated the way the authors stated the experimental hypothesis in a graphic manner. Also, I suggest to explicit the rationale of the study (i.e., why is it important? What are the issues to be taking into account/to be disentangled?). This could help the reader to follow the reason why the authors choose to perform the study. Another point concerns the use of references that are, for the vast majority, older than 5 years. I recommend the use of more recent works for this section.
Materials and Methods
The section 2.1. Procedures and Participants should heavily rewritten.
- Lines 133-134: are these sentences related to investigators? What does the sentence “Fill in at least 3 minutes, and no missing items can be Check the returned questionnaires one by one, and those with logical inconsistencies will be eliminated.” means? I suggest to divide the section by taking into account: the average time for questionnaire fulfilling; the time of the day for filling (if any); the order of the questionnaires; measures for preventing participants’ drop out;
- Lines 146-149: what kind of instruments did authors use to include/exclude volunteers?
Measures
All the rating scales are clearly and meticulously described. Authors clearly described the reliability index (Cronbach alpha) for Chinese translations of the aforementioned questionnaires
Statistical analyses
The section is well described and easy to follow. However, I suggest to include the computation of the effect size to estimate the magnitude of the effects. However, I suggest to better clarify the rationale for controlling the variable “gender” and “grade”, as the explanation given by the authors is shallow.
Results
This section appears clear and reader-oriented. However, I strongly recommend to avoid redundancy concerning contents present both in table and in the text. For what concern the p-value in the table, I suggest to report the exact value (for example 0.0002 instead of p < 0.001). As regard Figure 2 and Figure 3, I recommend to improve the quality of the figure (if is possible, I suggest the use of different colors to catch the eye of the reader). Another important point concerns the content of the captions; the way authors describe the contents of the figures has to be conceived as a stand-alone object (i.e., must be synthetic and exhaustive)
Discussion
The section 4.1 should be conceived in different way. First of all, the title does not identify the nature of the contents, and should indicate the direction of obtained results. The interpretation of the results appears superficial, and does not valorize the works that authors conducted. I suggest, to enhance the quality of the discussion, to discuss also the negative results. For example, how authors explain that “family dynamics was positively associated with all the big five personality traits except neuroticism”?
The same suggestions are recommended for subsection 4.2 and 4.3
For what concern the “limitations and implications” subsections, I appreciated the way the authors define al the limits they encountered, but I also suggest to describe ways to overcome them. Moreover, authors brilliantly describe practical implications of their works, and they shed further light of new potential
Reviewer 2 Report
- Lines 38-45, a number of sentences split by a semicolon start with a capital letter which is not proper English usage.
- Line 42 et al. usually ends with a period
- Lines 134-136 are too abbreviated and are not complete sentences or phrases.
- Lines 144-152 same issue as point #2.
- Line 193 I would like to see more evidence for the macro not needing normally distributed data, at least ideally; what evidence shows us that the macro is robust with respect to violations of normality?
- Line 194 I think the authors meant they were using Spearman rho correlations rather than a Spearman analysis; is it typical to use Spearman rho's rather than zero-order Pearson correlations for multivariate analyses?
- On line 200 gender and grade were deemed control variables but on line 218 only gender is mentioned. This inconsistency needs clarification.
- It is not clear why the other demographic variables were not used as controls as well as gender and grade.
- In Table 2, it is interesting that neuroticism is not substantially correlated with the other personality variables and that the other four personality variables are strongly intercorrelated.
- Line 214 why mention Table 3,4? It should be Tables 3 and 4 if both tables are meant.
- Figure 3 is of interest, of course, but did the components of family dynamics differ in terms of how the interaction came out?
- Line 274 mentions "moderately" and that makes this reviewer wonder if there were significant nonlinear, especially quadratic, relationships among the key variables at any point in the analyses?
- Lines 276-277 This does not appear to be a complete sentence.
- Line 305 College should have its first letter a capital letter.
- Line 333 What does "emotionally unreal" mean?
- References. The style is not consistent. In particular, for some references the first letter in each title is capitalized while in others only the first word's first letter.
- Some of the journal names are done with abbreviations and others not so much. Please ensure that the journal's style is followed consistently.
Round 2
Reviewer 1 Report
The Authors satisfactorily addressed all the comments and suggestions, and significantly improved the quality of the manuscript.
In order to be suitable for publication, I suggest minor issues to be taken into consideration.
- Authors should improve the quality of the figures, especially of figure 1.
- Authors should uniform the font type and size of axis description (I strongly recommend to increase the font size) between the figures.
- Authors should comment the magnitude of the effect size in the discussion section
Reviewer 2 Report
- The title should say "only-child status" rather than only child.
- Lines 33, 34 are not a complete sentence.
- Lines 89-92 are not a complete sentence.
- Lines 97-99 are confusing because on the one hand it appears neuroticism is the most important predictor while on the other it seems to be responsibility. Furthermore, it is not clear how "responsibility" ties into the big five personality traits - or does it mean something else entirely?
- Line 128 whether needs to start with a capital letter and since family type is the last phrase in the series it should be "and family type".
- Table 1, I'd suggest taking the headings for each variable and putting them in bold type and having the first word in the heading start with a capital letter. Thus, for example, you would have Location of the home or Family types. This should be done for all the headings, not just my examples, of course.
- Line 137, it would help to give a couple of examples of questions used in the SSFD.
- Line 154 The alpha for agreeableness is not "high" but "fair" at best.
- Line 168 It should be significance level rather than significant level.
- Line 276 It should be "are protective factors".
